# The High Temperature Co-Processing of Nickel Sulfide and Nickel Laterite Sources

**Robbie G. McDonald \*** and **Jian Li**

CSIRO Mineral Resources, PO Box 7229, Karawara, WA 6152, Australia; jian.li@csiro.au

\* Correspondence: robbie.mcdonald@csiro.au

**Abstract:** The pressure oxidation of low-grade nickel sulfide concentrate with high iron sulfides content generates significant amounts of sulfuric acid that must be neutralized. This acid can be utilized to leach metal values from ores such as nickel laterites. The present study demonstrates the use of a low-grade nickel concentrate generated from Poseidon Nickel Mt Windarra ore to enable additional nickel and cobalt extraction from a Bulong Nickel Operation nickel laterite blend. The co-processing of these materials at 250 °C, with oxygen overpressure, using total pulp densities of 30% or 40% w/w, and a range of nickel concentrate to nickel laterite mass ratios between 0.30–0.53, yielded base metal extractions of 95% or greater. The final free acid range was between 21.5–58.5 g/L, which indicates that enough in situ sulfuric acid was generated during co-processing. The acid was shown from mineralogical analysis to be efficiently utilized to dissolve the laterite ore, which indicates that the primary iron hydrolysis product was hematite, while the aluminum-rich sodium alunite/jarosite phase that formed hosts approximately 5% of the hydrolyzed iron.

**Keywords:** nickel laterite; nickel sulfide; high pressure acid leaching; co-processing; pressure oxidation; QXRD analysis

---

## 1. Introduction

The processing of nickel laterites by high pressure acid leaching (HPAL) typically uses feeds with an average nickel content not less than 1.1%–1.2% (cut-off grade 1.0%) [1] though recently several nickel laterite projects have been revalued with lower nickel cut-off grades based upon the cobalt content of the ore. It may also be possible to revalue a low-grade nickel laterite deposit if the grade of the ore feed that is processed can be enhanced by the addition of a supplementary nickel-containing material, also referred to as co-processing [2] or combined pressure acid leaching [3]. In Western Australia such materials are abundant, may be in proximity and include sources such as oxidized zones of sulfide ore bodies, low-grade sulfide ores and concentrates, and nickel matte leach residue. Depending upon the material chosen, there is also potential to generate sulfuric acid from the oxidation of sulfide minerals, which in addition to the extraction of more metal values will impact the operating costs for the co-processing approach.

The concept of using elemental sulfur or a sulfide-containing material to generate in situ sulfuric acid to enable the leaching of a nickel laterite ore is not novel [4]. Although the roasting of sulfur or pyritic ores with nickel laterites dates back many decades, with a proposed improvement for example being described by the invention of Kichline [5], the hydrometallurgical treatment of nickel laterites using pyrite by O'Neill [6] appears to be the first instance of such an oxidative leaching approach. Subsequently, however, there have been few scientific studies on this topic [3,7], in addition to a handful of patent applications [8–13]. This is presumably because sulfur burning to generate sulfuric acid for nickel laterite leaching is well established, while the reliable sourcing of a suitable, readily accessible, consistent grade sulfidic ore/concentrate source may be problematic. Certainly,

the hydrometallurgical co-processing of nickel laterites with a sulfidic resource, while potentially conferring several benefits, also introduces issues that must be appropriately addressed in the design and operation of a commercial plant.

It is widely acknowledged that HPAL (e.g., [14,15]), whilst being a proven technology, has been employed in projects that are:

(1)　high in capital expenditure and, for several operations, there have been significant cost blow outs;
(2)　in most cases have required long ramp-up times, often not achieving nameplate capacity;
(3)　best suited to limonitic fractions of the nickel laterite profile to ensure low acid consumption;
(4)　composed of variable and complex downstream processing operations; and
(5)　large generators of waste materials that must be treated and stored appropriately.

The perceived advantages of co-processing (as the leaching operation) where most, if not all, of the sulfuric acid is generated in situ include:

(1)　The potential to inject a higher pulp density ore/concentrate mix enhancing throughput, as it has been shown that the addition of nickel sulfide-containing material to nickel laterite ore lowers the pulp yield stress [16];
(2)　Little or no requirement to burn sulfur to generate sulfuric acid, with the elimination of potential attendant issues for a sulfuric acid plant such as the inadvertent generation of hydrogen;
(3)　Less loss of heat from sulfur burning not recovered as steam or hot water that can be used in the process along with potential lower plant use of water for cooling in the sulfuric acid plant;
(4)　The in situ generation of heat resulting in reduced requirement for steam injection and associated increase in working pulp density;
(5)　Depending upon the sulfidic ore/concentrate source, the generation of a higher metal value concentration leach liquor, e.g., pyrrhotite, which generally contains low but not insignificant amounts of nickel, does not need to be rejected when generating a nickel concentrate;
(6)　Greater tolerance for high magnesium content in the nickel concentrate compared with material sent to nickel smelters, though preferably hosted by poorly leached minerals such as talc;
(7)　The in situ solubilization of toxic impurities, such as arsenic, which can then be more readily managed via the formation of stable ferric arsenate compounds.

In contrast, several matters would need to be addressed in a commercial co-processing operation:

(1)　If sulfur is not burned, there needs to be an alternative energy source available to generate steam and/or electricity requirements;
(2)　An oxidant is required for in situ generation of the sulfuric acid, which is expected to be oxygen that is generated on site;
(3)　The complexity of the equipment will be greater than that required for high pressure acid leaching and, if oxygen is used as the oxidant, some of the materials of construction may be more exotic/expensive.

To date there have been few contributions to the topic of nickel laterite plus sulfidic nickel ore/concentrate co-processing [3,7]. The pilot scale study of Quinn et al. [3] employed 70:30 nickel laterite/sulfide blends having low nickel (1.24%–1.63%) and sulfide (3.2%) contents milled in previously generated heap leach liquor. These blends were leached at 220 °C using 800–1000 kPa oxygen overpressure and supplementary sulfuric acid addition of 150 kg/t blended ore. In comparison, Ferron and Fleming [7] conducted batch test work at higher temperature, 250 °C, using 690 kPa oxygen overpressure with additions of elemental sulfur, pyrrhotite (containing 0.94% Ni) or low-grade copper sulfide concentrate (containing 3.05% Ni) as the sulfur-bearing materials.

The present study was undertaken using a nickel laterite blend generated (prior to closure in 2003) at the Bulong HPAL plant together with a low-grade nickel concentrate (8.1% Ni) having high iron and

total sulfide contents provided by Poseidon Nickel Limited. The primary aim was to demonstrate the technical viability for co-processing these materials, from which information relating to the metals' extraction performance and changes in the mineralogical composition was obtained.

## 2. Materials and Methods

Run-of-mine Bulong nickel laterite blended ore, taken from that fed to the now-closed Bulong Nickel Refinery, Western Australia, was wet screened to −500 µm and filtered to a moist cake to provide the feed material used in test work. Poseidon nickel concentrate (received dry) was screened to −75 µm to generate the sulfidic feed material. The site process water used in these tests had the following composition (mg/L): $Mg^{2+}$ (210), $Na^+$ (1920), $Ca^{2+}$ (70), $K^+$ (50), $SO_4^{2-}$ (560), and $Cl^-$ (3300).

Leaching of individual feeds and blends was conducted at 250 °C using a 1 gallon Grade 3 titanium Parr autoclave (Parr Instrument Company, Moline, IL, USA) with a dual pitched blade impeller driven at a speed of 750 rpm in site water from the Mt. Windarra nickel project. The leaching of the nickel laterite blended ore used a pulp density of 30% w/w whereas blends of nickel laterite and nickel concentrate generally employed a pulp density of either 30% or 40% w/w. Nickel laterite leaching was commenced by injecting the required amount of (concentrated) sulfuric acid into the autoclave, the process referred to as HPAL. Tests in which nickel concentrate was also present (co-processing) were conducted with various laterite to concentrate ratios, without sulfuric acid addition. These tests were started via the continuous injection of industrial oxygen to a set overpressure in the range 100–250 kPa, into a head space previously purged with industrial nitrogen. The rate of oxygen flow into the autoclave head space was controlled using a Teledyne Hastings Instruments Model HFC-D-302 Flowmeter/Controller with Power[POD] 400 Power Supply/Totalizer (Hampton, VA, USA). Experiments were run for either 90 min (HPAL) or 120 min (co-processing). The rate of oxygen flow was in the range of 2.8–4.0 g/min.

It is noted here that a gas entrainment impeller was trialed as an alternative to the dual impeller system but under the conditions used of low head space pressure and fixed maximum oxygen flow rate, it only had a small impact upon the rate of oxygen usage as the reaction was nearing completion. Otherwise, it had no impact on the rate of oxygen usage under the conditions used.

The methods used for kinetic sampling, sample preparation and elemental analysis, free acid and ferrous iron determinations, X-ray diffraction (XRD) sample preparation and measurement, and Quantitative X-ray Diffraction (QXRD) analysis have been previously described in detail [17].

## 3. Results and Discussion

### 3.1. Feed Materials

Laterite Ore, Nickel Concentrate, and Site Process Water

The elemental compositions of the feed materials (after drying) used in the test work are given in Table 1. These data indicate (in West Australian terms) a low-grade nickel concentrate (8.1% Ni) and a high-grade nickel laterite blend (1.9% Ni); the moisture content of the laterite blend was 28.7% w/w.

**Table 1.** Elemental compositions of the low-grade nickel concentrate and high-grade laterite feeds employed in this study (% w/w).

| Feed | Ni | Co | Cu | Mg | Mn | Fe | Al | Cr | Ca | Na | Si | S |
|------|------|-------|-------|-------|-------|------|------|-------|-------|-------|------|------|
| Concentrate | 8.11 | 0.138 | 1.28 | 0.651 | 0.037 | 45.0 | 0.318 | 0.175 | 0.421 | 0.082 | 2.48 | 30.3 |
| Laterite | 1.90 | 0.131 | 0.011 | 3.32 | 0.456 | 24.6 | 2.47 | 0.857 | 0.170 | 0.201 | 16.8 | 0.02 |

The nickel concentrate (moist when received) was composed largely of sulfide minerals that when exposed to air were mildly oxidised. The mineralogical composition of the concentrate after pre-conditioning in the autoclave, sampling, and preparation for QXRD analysis is given in Table 2.

**Table 2.** Mineral compositions of the blended nickel laterite ore and (hydrothermally heated) fresh nickel concentrate feeds employed for the current test work.

| Bulong Laterite Blend | % (w/w) | Nickel Concentrate | % (w/w) |
|---|---|---|---|
| Nontronite | 55 | Pyrrhotite 4C | 39 |
| Goethite | 26 | Pyrite | 13 |
| Spinel * | 7 | Pentlandite | 8 |
| Maghemite | 2 | Violarite | 9 |
| Magnesite | 1 | Chalcopyrite | 3 |
| Quartz | 3 | Quartz | 4 |
| Clinochlore | 3 | Clinochlore | 3 |
| Actinolite | 1 | Actinolite | 2 |
| Lizardite | 0.7 | Talc | 4 |
| Talc | 0.3 | Hematite | 4 |
| | | Hydronium jarosite | 5 |
| | | Nickel hexahydrite | 1.8 |

\* Magnetite/chromite/magnesioferrite.

The Bulong nickel laterite ore is essentially a blend of limonite and smectite zone fractions. QXRD analysis presented in Table 2 indicated the mineral composition to be 26% goethite, 55% nontronite, 7% spinel minerals, 2% maghemite, 3% clinochlore, and 3% quartz, along with several other minor minerals that include an amphibole mineral (possibly actinolite). Here the nontronite content was determined using the model published by Wang et al. [18]. A sample of Bulong nontronite characterized in greater detail was found to contain significant amounts of Ni (2.0%), Mg (2.1%), Fe (17.8%), and Al (5.4%) [19]. The estimated c-axis dimension of 3.016 Å deviates from that of pure goethite, 3.023 Å [20,21], which is consistent with this phase being a host for various metals that include, but are not limited to, Ni, Co, and Al.

*3.2. Nickel Laterite Leaching*

The leaching of the Bulong nickel laterite blend under HPAL conditions, which employed an acid addition of 364 kg/t dry ore, was rapid as shown in Figure 1. High extractions of the main metals of interest were obtained within 10 min, though extractions thereafter continued to increase slowly, for nickel and cobalt reaching ~95% after 90 min. Conversely, the extractions of metals such as iron and aluminum were low and the final concentrations for these metals were 1.1 g/L and 0.36 g/L, respectively. Although not shown in Figure 1, the extraction of chromium was also low and gave a final concentration of 0.04 g/L. The free acidity, which started at ~150 g/L, also dropped rapidly during the first 10 min and slowly decreased thereafter as the metal extractions were optimized and the residue solids equilibrated with respect to the leach liquor. This behavior is quite typical for batch HPAL tests and is discussed further below.

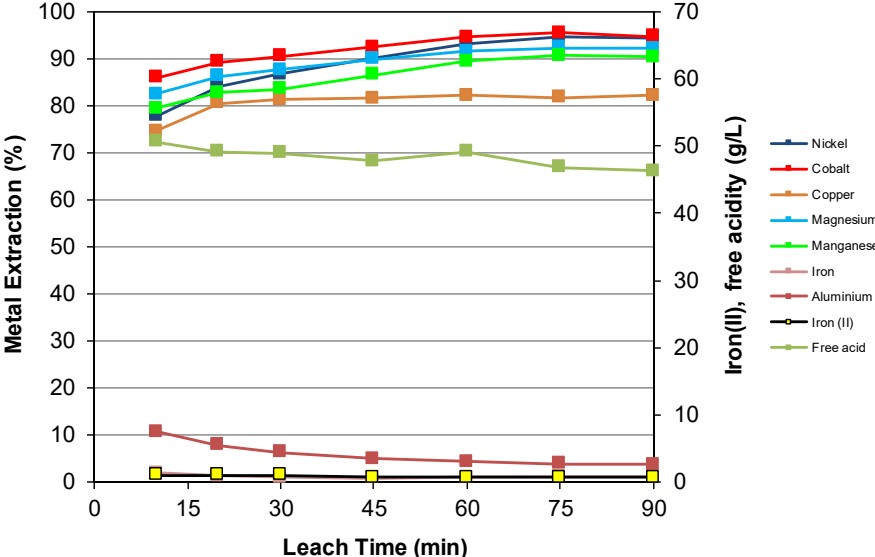

**Figure 1.** Metal extractions, ferrous iron concentration and free acidity for samples taken during the high pressure acid leaching (HPAL) of 30% w/w blended nickel laterite ore in site water at 250 °C using 364 kg/t dry ore acid addition.

Although most of the leaching was completed within the first 10 min, the mineralogy of samples taken at that time and over the remainder of the test provide some interesting insights. Close examination of the changing XRD patterns reveals first, in Figure 2A, the collapse in intensity of the nontronite (001) peak centered at ~6.5° 2θ due to leaching/acid modification. The resistance of talc to leaching is characterized by the endurance of the (001) peak centered at ~11° 2θ, and the slow leaching of other aluminosilicates, here lizardite and clinochlore with characteristic (001) and (002) peaks at 14.1° and 14.3° 2θ, respectively. The leaching of goethite can be visually identified from the (110) peak at 24.8° 2θ in combination with other peaks shown in Figure 2B, such as the (021) and (111) peaks at 40.6° and 42.9° 2θ, respectively. Maghemite (and spinel minerals) also dissolve slowly as evidenced primarily by the peak at 35.3° 2θ.

It is notable in Figure 2A that a broad hump centered at ~25° 2θ due to the formation of poorly crystalline ("amorphous") silica overlaps a region where diffraction intensity due to nontronite and goethite occur in the feed sample. Visually, the (001) nontronite peak and peaks assigned to goethite have all but disappeared in the t60 residue sample though the remaining hump still contains (apart from a sharp peak due to quartz) several bumps at ~23.0°, ~23.8°–24.0° and ~25.2° 2θ. These are expected to indicate to partial crystalline transformation of the poorly crystalline silica. More specifically, it has been well documented that hydrothermal conditioning of amorphous silica formed from the leaching of silicate minerals, including nontronite, occurs. Amorphous silica under various conditions of pH and temperature can be transformed to (α-)cristobalite (e.g., [22–26]), silica-K (keatite) [25,26], and quartz (e.g., [22,24–26]) where cristobalite and silica-K have been indicated as precursors to the formation of quartz [25,26]. As crystallization is facilitated in alkaline medium [27–29], the broad nature of the bumps observed here seems to be consistent with the low pH of the leach liquor. The bump at ~23.0° 2θ is present in the XRD pattern for keatite (though this may be also due to partially leached nontronite), that at ~25.2° 2θ occurs in the pattern for α-cristobalite, and while those in the range ~23.8°–24.0° 2θ are seen in the XRD pattern for Opal-CT [25,30] and have be previously assigned to tridymite [23]. However, tridymite is only known to form at temperatures above 750 °C [23,31], while the XRD pattern for nanocrystalline cristobalite prepared in neutral solutions at 200 °C has a shoulder that occurs in a similar position.

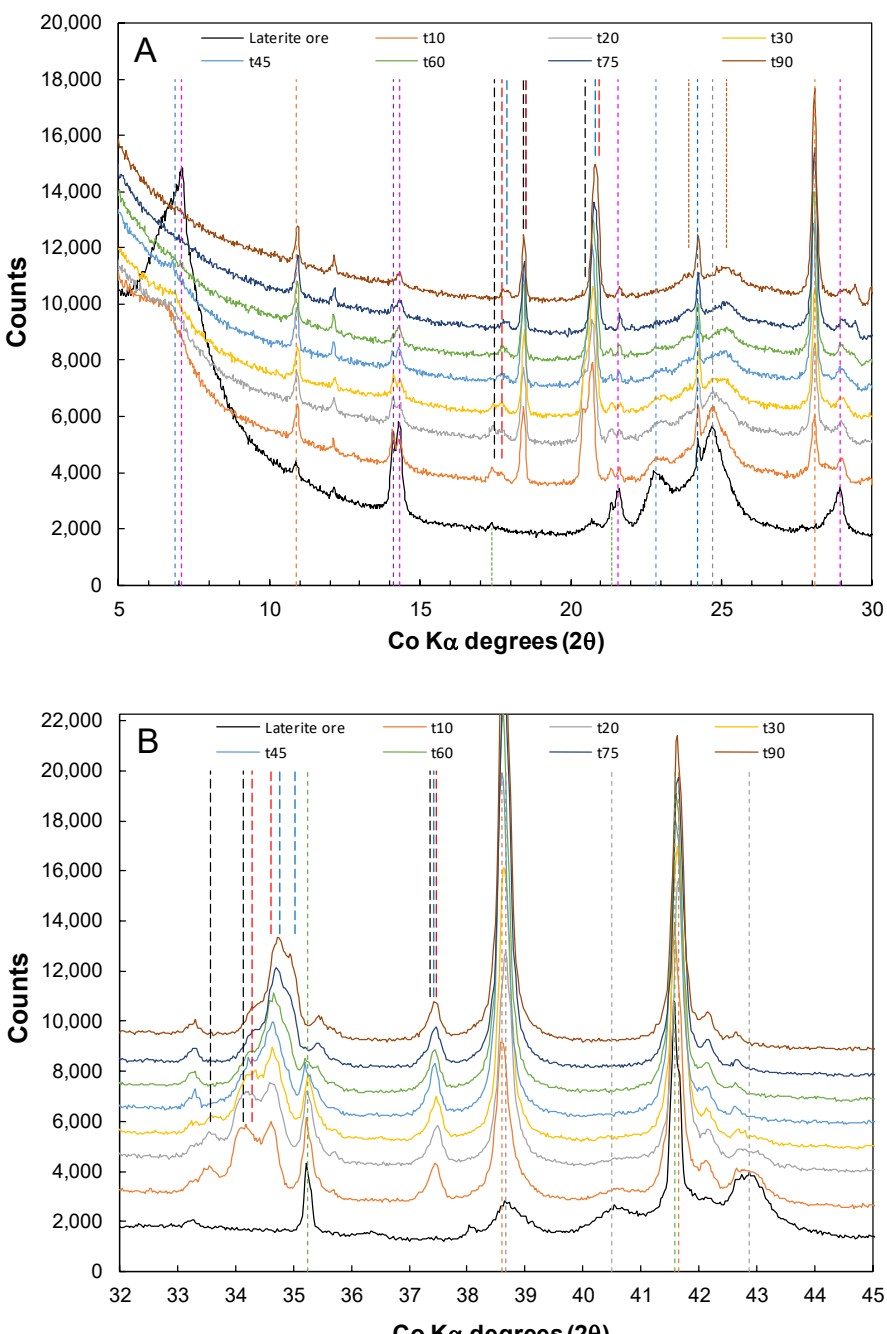

**Figure 2.** XRD patterns for selected angle ranges of the feed ore and samples (t10, t20, t30, t45, t60, t75, and t90) taken during the HPAL of 30% w/w blended nickel laterite ore in site water at 250 °C using 364 kg/t dry ore acid addition. Selected peaks positions (° 2θ) are shown for the following minerals: nontronite (6.85, 22.82), clinochlore (7.07, 14.33, 21.57, 28.94), talc (10.90), lizardite (14.12), quartz (24.22), goethite (24.70, 38.67, 40.49, 42.86), maghemite (17.39, 21.43, 35.25, 41.58), hematite (28.10, 38.60, 41.65), Na-jarosite (17.45, 18.41, 20.50, 33.56, 34.13, 37.36), Na-alunite/jarosite (17.72, 18.52, 20.93, 34.28, 34.62, 37.46), and Na-alunite (17.90, 18.50, 20.83, 34.76, 35.03, 37.43). (**A**) angle range 5–30° 2θ; (**B**) angle range 32–45° 2θ.

The first sample collected after 10 min indicates the presence of at least two alunite/jarosite phases, as evidenced by the profile which shows (012) peaks at 20.5° and 20.8° 2θ (Figure 2A). A more complex scenario is revealed by the profile that contains the (021) and (113) peaks in the range 33.5°–35.5° 2θ, and which evolves with time (Figure 2B). For these profiles the shifts of the (012), (021) and (113)

peaks to larger angle are consistent with shortening of the unit cell a-dimension. In comparison the c-dimension remains relatively unchanged as shown for the (006) peaks at ~37.4° 2θ. That there appear to be multiple alunite and/or jarosite phases produced is consistent with the ability of these phases to form solid solutions with formula $AB_3(SO_4)(OH)_6$ in which the A and B cation sites can be occupied by one of more cations simultaneously, (A = $H_3O^+$, $Na^+$, $K^+$, $NH_4^+$ etc.) and (B = $Fe^{3+}$, $Al^{3+}$, $Cr^{3+}$ etc.), although there are limits to the mutual solubility of multiple cations in the solid solutions [32,33]. Rietveld refinement here employed four alunite/jarosite models from which the average composition of the alunite/jarosite phases were calculated from parametric expressions like those derived for the a and c unit cell lengths of alunite/jarosite solid solutions by Makhmudov and Kashkai [34] and utilized by Whittington [35]. These expressions are given as follows:

$$a = 5.549 + 0.0136r_A + 2.46r_B + 0.556r_X \tag{1}$$

$$c = 15.473 + 1.488r_A - 0.704r_B \tag{2}$$

where $r_A$ and $r_B$ represent averaged ionic radii for the monovalent and trivalent cation sites, respectively, while $r_X$ is the ionic radius of the S(VI) ion. Ionic radius values (Å) used were $Na^+$ (1.09), $K^+$ (1.49), $H_3O^+$ (1.37), $Fe^{3+}$ (0.645), $Al^{3+}$ (0.51), and $S^{6+}$ (0.29), which in some cases differ marginally from values published by Shannon [36]. Representing alunite/jarosite phases containing these cations by the general formula $K_mNa_n(H_3O)_{1-m-n}Fe_pAl_{3-p}(SO_4)_2(OH)_6$, and with R representing the ionic radius, the values for $r_A$, $r_B$ and $r_X$ were calculated as follows:

$$r_A = mR(K^+) + nR(Na^+) + (1 - m - n)R(H_3O^+) \tag{3}$$

$$r_B = pR(Fe^{3+}) + (3 - p)R(Al^{3+}) \tag{4}$$

$$r_X = R(S^{6+}) \tag{5}$$

The above model was fitted to a more extensive set of unit cell length and chemical composition data [37–52] by minimizing the differences between the actual and calculated unit cell values using the Microsoft Excel Solver function. The following modified equations were derived for use in this study:

$$a = 5.504 - 0.0060r_A + 2.59r_B + 0.544r_X \tag{6}$$

$$c = 15.214 + 1.659r_A - 0.712r_B \tag{7}$$

From these empirical expressions the (Na, $H_3O$) and (Fe, Al) occupancies in hydronium/natro-alunite/jarosite solids formed were estimated and used during the Rietveld analyses. The picture revealed by the modelling is that initially sodium-and iron-rich jarosite and sodium-rich, mixed alunite/jarosite phases are formed. As the reaction proceeds, the iron-rich jarosite disappears and is replaced by a sodium-rich alunite phase which has smaller unit cell a-dimension. Overall, the alunite/jarosite phases are predicted from the QXRD analysis and compositional modelling to become richer in sodium during the first 30 min (increasing from 0.65% to 0.81%) and remain near constant in sodium content thereafter. In comparison these phases become richer in aluminum over the entirety of the reaction (increasing from 1.27% to 2.47%). Such behavior during the batch HPAL of nickel laterite ores has previously been reported in other studies [53] and occurs according to the following generalized equation:

$$Na_x(H_3O)_{1-x}Fe_yAl_{3-y}(SO_4)_2(OH)_6 + aNa_2SO_4(aq) + bAl_2(SO_4)_3(aq) \rightarrow$$
$$Na_{x+2a}(H_3O)_{1-x-2a}Fe_{y-2b}Al_{3-y+2b}(SO_4)_2(OH)_6 + aH_2SO_4(aq) + bFe_2(SO_4)_3(aq) + aH_2O \tag{8}$$
$$(a \leq 1 - x, b \leq y)$$

The small increase in total sodium, compared with the significant increase in total aluminum, contents in the residue, is consistent with depletion of both these elements in the leach liquor during

the reaction. Final concentrations of 120 mg/L Na and 360 mg/L Al were reached. These observations are also consistent with thermochemical calculations indicating that (1) natrojarosite is only marginally more stable than hydronium jarosite [41] and (2) the stability of natroalunite/jarosite solid solutions is greatly enhanced as the degree of aluminum substitution increases [42].

Although nickel and cobalt extractions were 94%–95% during the HPAL of the Bulong laterite blend, further leaching of these elements should be attainable. It is expected that the small fraction of these metals (0.13% Ni and 0.009% Co in the final residue) not extracted are hosted by incompletely leached minerals that include clinochlore, lizardite, and spinel minerals; more complete extraction would be facilitated by leaching for a longer period and/or using a higher acid addition. That said, the final free acidity of 46 g/L indicated that the acid addition of 364 kg/t dry ore was probably suitable, given that HPAL plants historically have targeted a free acidity of at least 50 g/L (e.g., [54]). While such targets have typically been established from deriving the relationship between nickel extraction and free acidity (e.g., [55]), it has been known for some time that such relationships are laterite sample dependent [56]. Rather, the "at temperature" acidity, i.e., [H$^+$] under the operating conditions used, has been demonstrated to be a better indicator and a value of above 0.1 M was indicated to be enough to obtain nickel extractions above 95% [56]. Given that this target extraction was reached in the current study, it is expected that the "at temperature" acidity was also sufficiently high.

### 3.3. Nickel Sulfide with Nickel Laterite Co-Processing

Instead of supplying the sulfuric acid, tests were completed to demonstrate that all of the acid required could be generated in situ to leach the laterite component of nickel sulfide concentrate/nickel laterite blends. The amount of sulfuric acid generated by pressure oxidation of the nickel sulfide concentrate was previously estimated to be ~800 kg/t concentrate [17].

Several tests were conducted using varying ratios of nickel sulfide concentrate to nickel laterite (and calculated sulfuric acid generated per tonne dry laterite ore): (1) 7.0:23.0 (247 kg/t), (2) 10.5:19.5 (437 kg/t), and (3) 12.0:28.0 (370 kg/t). For a nickel sulfide to nickel laterite ratio of 7.0:23.0, the extraction data are shown in Figure 3. Noticeably, the nickel, cobalt, and copper extractions reached ~95% while manganese and magnesium were marginally lower; oxygen injection was essentially complete after 45 min. Most of the magnesium not extracted was associated with incompletely leached clinochlore and poorly leached talc. Initially, noticeable amounts of magnesium and manganese reported to the t0 sample (taken after the heating period). Ion exchange of the magnesium ions located in the nontronite interlayer enables this metal to enter the leach liquor and explains the magnesium extracted at this time. Furthermore, reduction of manganese (IV), expected to be present in the limonite component of the Bulong laterite blend, by galvanic coupling with sulfide minerals [57] and/or the ferrous iron released from surface oxidation products of sulfide minerals, results in solubilization of manganese (II). Notably, the lag in cobalt compared to nickel extraction is not as significant as for the pressure oxidation of the nickel sulfide concentrate. This is because 64% of the cobalt is associated with the laterite component from which it is more readily leached. This also infers that a significant proportion of the cobalt in the nickel laterite blend is associated with manganese oxide minerals as noted in previous studies [58,59]. That there is a lag in the leaching of copper is like that during pressure oxidation of the concentrate [17] (also given the copper content of the laterite is minimal at ~0.01%). Also, like pressure oxidation it is also expected that the pyrite and associated cobalt in this component will be the last sulfide mineral to be completely (or nearly so) leached [17]. The addition of a low relative amount of concentrate results in a low maximum ferrous iron concentration being reached after 30 min. Thereafter, oxidation of the ferrous iron and hydrolysis of the ferric iron results in the net generation of sulfuric acid. Some of this acid is subsequently used since leaching of the laterite component continues to occur until the conclusion of the reaction while the free acidity drops from the peak value after 45 min.

At a higher concentrate to laterite ratio of 10.5:19.5, the extents of nickel, cobalt, and copper extraction are marginally greater, all reaching ~98% (Figure 4). Metal extractions follow similar profiles to those seen previously, though with the larger concentrate to laterite ratio, the initial extraction of

manganese is greater. Furthermore, there is a higher peak concentration of ferrous iron and more sulfuric acid production per unit mass of laterite. This leads to a higher final free acidity, facilitates greater metal extractions, and results in a higher final extraction of iron.

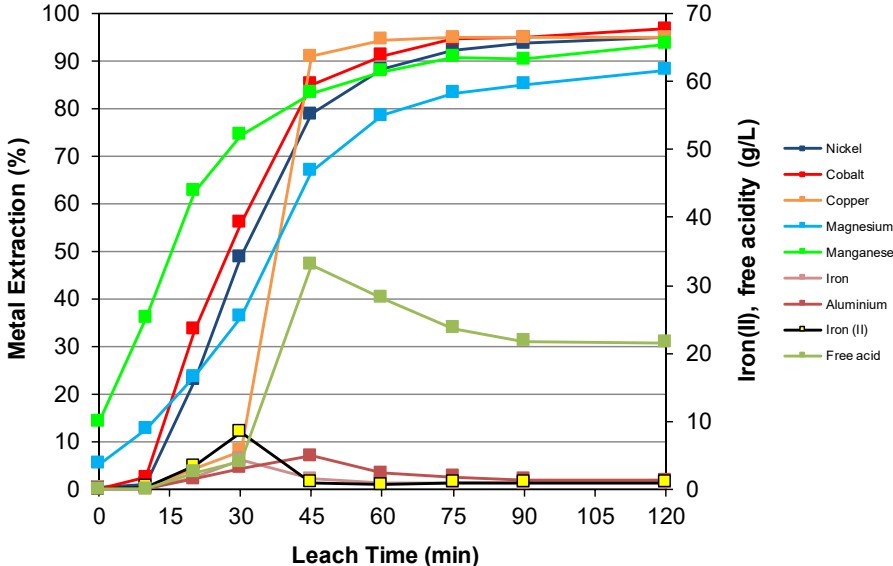

**Figure 3.** Metal extractions, ferrous iron concentration, and free acidity for samples taken during the co-processing of 30% w/w total, 7% nickel concentrate with 23% nickel laterite ore, in site water at 250 °C with ~150 kPa $O_2$ overpressure.

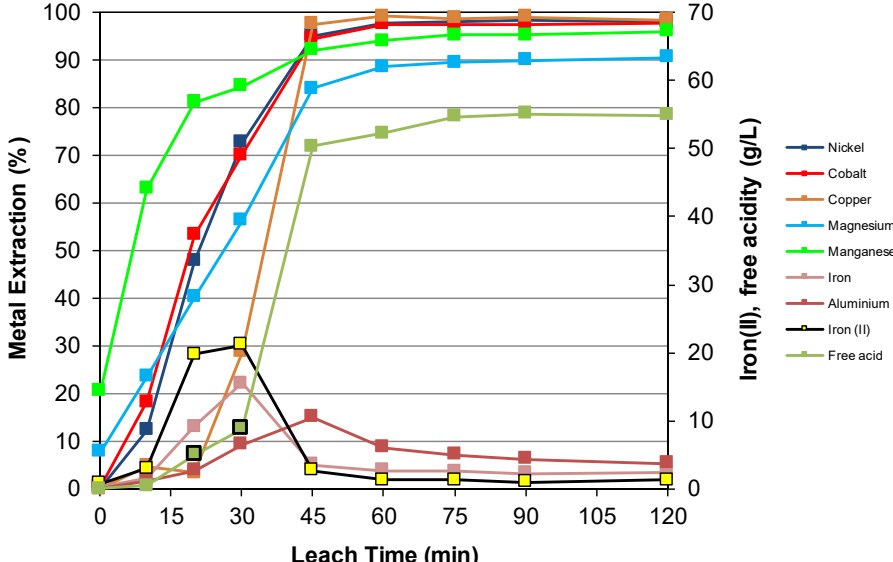

**Figure 4.** Metal extractions, ferrous iron concentration, and free acidity for samples taken during the co-processing of 30% w/w total, 10.5% nickel concentrate with 19.5% nickel laterite ore, in site water at 250 °C with ~100 kPa $O_2$ overpressure.

The concentrate to laterite ratio of 12.0:28.0 is intermediate to that used in previous tests but also resulted in nickel/cobalt extractions of ~98% and copper extraction of ~97% (Figure 5). In comparison, the reaction was slower with oxygen consumption virtually completed after 60 min rather than 40 min as for the previous tests. Furthermore, the rate of oxygen consumption was less than the set flow rate, indicating that oxygen uptake was limited by mass transfer to and within the reacting slurry (with the oxygen solubility determined by the oxygen partial pressure of ~250 kPa). The average oxygen consumption from all three tests was 0.58 grams per gram of concentrate.

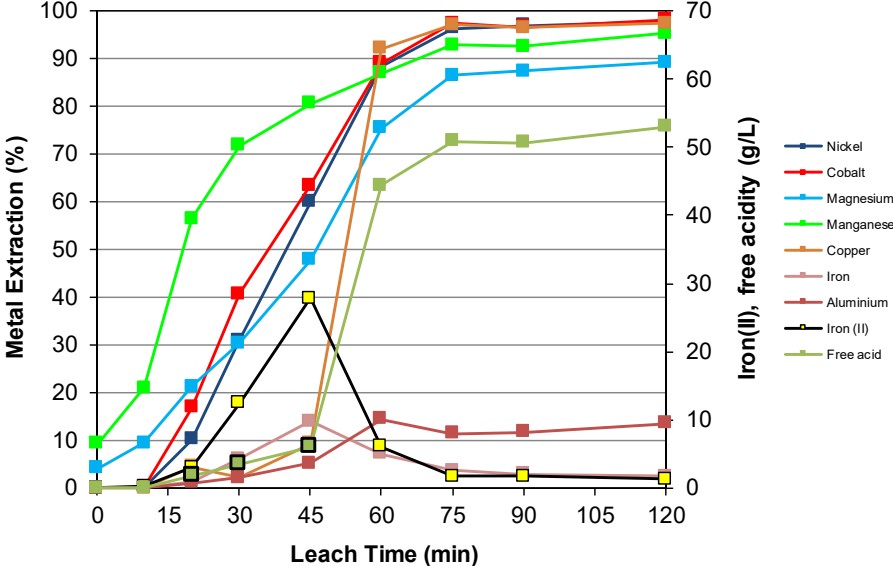

**Figure 5.** Metal extractions, ferrous iron concentration and free acidity for samples taken during the co-processing of 40% w/w total, 12.0% nickel concentrate with 28.0% nickel laterite ore, in site water at 250 °C with ~250 kPa $O_2$ overpressure.

For all co-processing tests, base metal extractions of 95% or greater were obtained, indicating that a wide range of concentrate to laterite mass ratios, here 0.30–0.53, can potentially be used to effect good metal value extractions. The ranges of concentrations produced in these tests were (in g/L): Ni 15.1–23.0, Co 0.64–0.90, and Cu 1.4–2.3. In comparison, the impurities including those requiring subsequent removal reported to the leach liquor with the following concentration ranges (in g/L): Fe 1.7–5.1, Al 0.2–1.7, Cr 0.06–0.20, Mn 1.6–2.3, and Mg 11.2–16.2; the final free acid range was 21.5–58.5 g/L.

Selected sections of the XRD pattern for the feed and residue samples taken during the coprocessing of 19.5% w/w nickel laterite ore with 10.5% w/w nickel concentrate are shown in Figure 6. The corresponding QXRD analysis data are shown in Figures 7–9 where the normalized mineral content represents the percentage of minerals relative to the solids amount used in the reaction. This means that the absolute change in a mineral's content can be shown.

In the co-processing system, the primary peak due to nontronite persists up to the t30 sample and has completely disappeared thereafter (Figure 6A). Although the pH of the leach liquor is quite low, 0.88, it is only after the ferrous iron concentration drops and a significant amount of hematite is formed (with co-generation of sulfuric acid) between 30 and 45 min that disintegration of the clay mineral silicate framework commences to any significant extent. It is noted here, however, that the composition of the nontronite is expected to change during the first 30 min, with exchange of cations between the leach liquor and both structurally bound and interlayer sites in the nontronite occurring. As Bulong nontronite samples are significant hosts for aluminum, magnesium, and iron [19], leaching of this phase releases all these elements which can all be ion exchanged before and during structural breakdown caused by the leaching process. The surge in aluminum extraction when the nontronite structure breaks down leads to the formation of aluminum-rich sodium alunite/jarosite, which becomes marginally more enriched in sodium and aluminum by the conclusion of the reaction, as discussed previously, consistent with the shift of peaks to larger diffraction angles. The steady increase in magnesium extraction (Figure 4) suggests not only that magnesium-containing minerals such as clinochlore and lizardite are leaching but this metal has also been ion exchanged from the nontronite structure. QXRD analysis suggests there is enough talc remaining to host ~10% of the magnesium and this largely accounts for the unleached magnesium, as noted previously.

The XRD data in Figure 6 show that pyrrhotite is not present in samples taken after t0. This is consistent with its behavior during pressure oxidation and hydrothermal replacement by pyrite and

marcasite [17], the latter being identified in the t10 residue. Similarly, pentlandite is not present in the t10 residue indicating that it is rapidly oxidized.

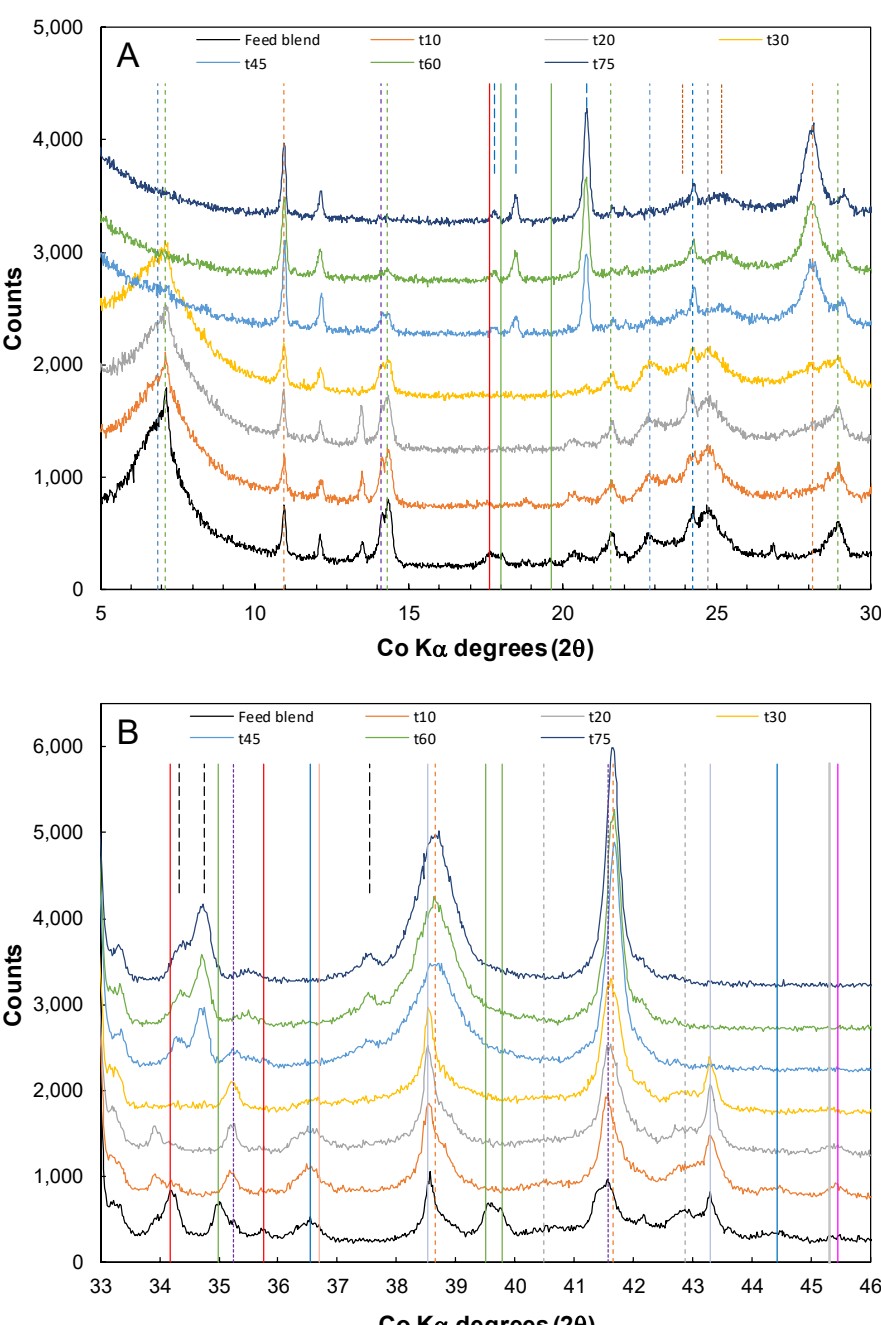

**Figure 6.** XRD patterns for selected angle ranges of the feed blend after heating to temperature (t0) and samples (t10, t20, t30, t45, t60, and t90) taken during the co-processing of 19.5% w/w nickel laterite ore with 10.5% w/w nickel concentrate in site water at 250 °C. Selected peaks positions (° 2θ) are shown for the following minerals: nontronite (6.85, 22.82), clinochlore (7.12, 14.33, 21.57, 28.94), talc (10.95), lizardite (14.12), quartz (24.22), goethite (24.70, 38.67, 40.49, 42.86), pyrrhotite 4C (18.01, 19.64, 34.98, 39.51, 39.78), pyrite (38.52, 43.29), marcasite (45.45), pentlandite (17.63, 24.18, 34.18, 35.75), violarite (36.54, 44.43), vaesite (36.70, 45.30), hematite (28.10, 38.60, 41.65), Na-alunite (17.80, 18.49, 20.80, 34.33, 34.75, 37.55). (**A**) angle range 5–30° 2θ; (**B**) angle range 33–46° 2θ.



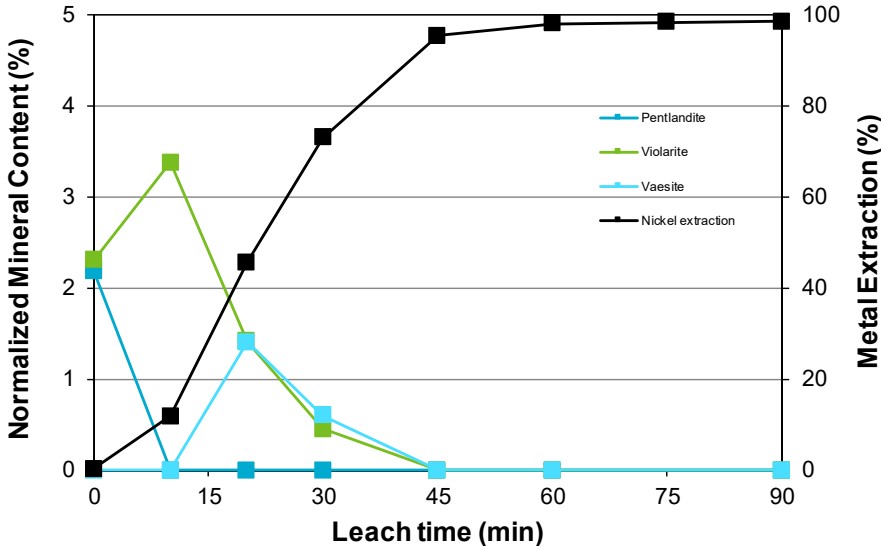

**Figure 7.** Changes in the composition of nickel sulfide minerals in the leach residue with corresponding nickel extraction for samples taken during the co-processing of 19.5% w/w nickel laterite ore with 10.5% w/w nickel concentrate in site water at 250 °C.

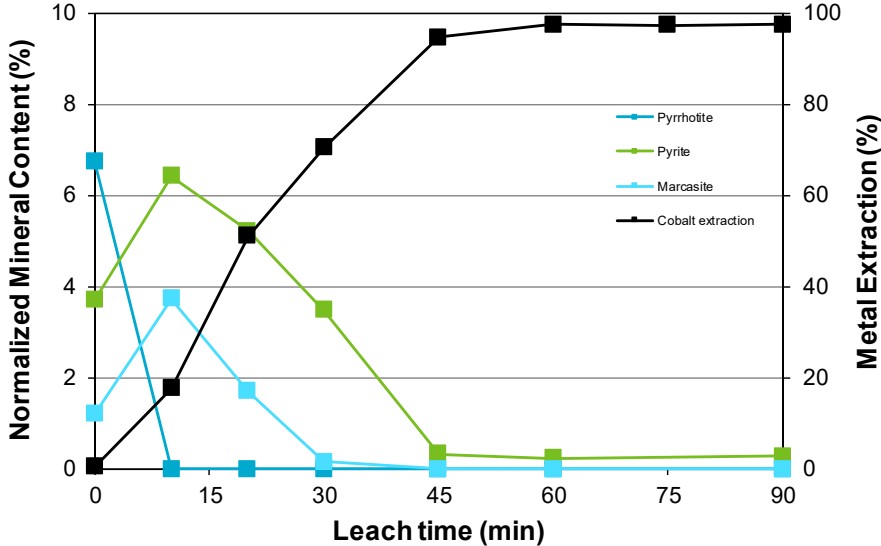

**Figure 8.** Changes in the composition of iron sulfide minerals in the leach residue with corresponding cobalt extraction for samples taken during the co-processing of 19.5% w/w nickel laterite ore with 10.5% w/w nickel concentrate in site water at 250 °C.

The QXRD data presented in Figures 7 and 8 demonstrate similar patterns for the changes in nickel sulfide and iron sulfide mineralogy as a function of leaching time during co-processing to those observed for the pressure oxidation of the low-grade nickel concentrate [17]. It is notable in Figure 8 that pyrite is the most refractory sulfide and though in the co-processing system cobalt extraction more closely follows nickel extraction. As previously discussed, 64% of the cobalt content is associated with nickel laterite minerals, hence the current observation suggests that metal is more sustainably extracted from the laterite minerals, and more particularly from manganese oxides associated with the goethite-rich component of the blend, than from the sulfide minerals. In comparison, only 30% of the nickel is associated with the nickel laterite minerals and the gradual increase in nickel extraction is largely sustained by the leaching of sulfide minerals.

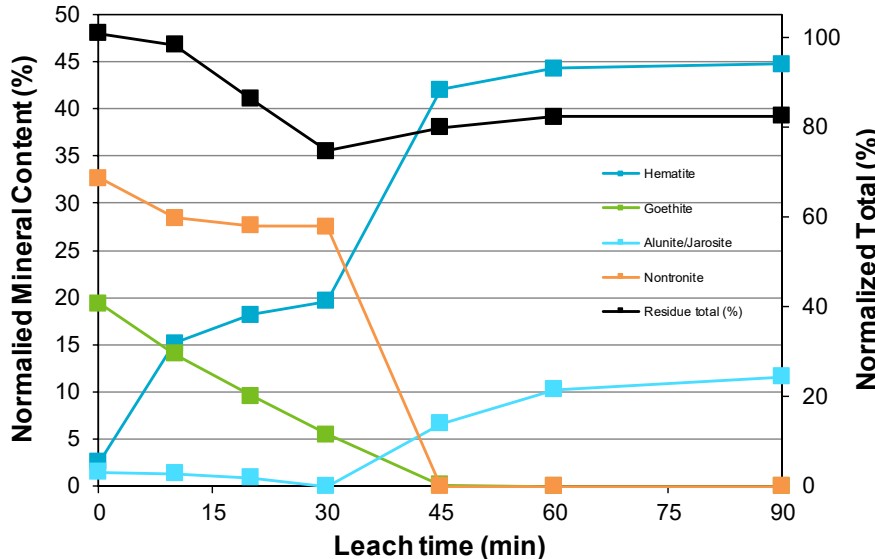

**Figure 9.** Changes in the composition of iron hydrolysis products in the leach residue for samples taken during the co-processing of 19.5% w/w nickel laterite ore with 10.5% w/w nickel concentrate in site water at 250 °C.

The behavior of the primary iron-containing oxide minerals shown in Figure 9 indicates a gradual decrease in the goethite content with leaching time. In comparison, the hematite content increases rapidly during the first 10 min, presumably consistent with the hydrothermal transformation of pyrrhotite to pyrite and marcasite, associated release of ferrous iron for subsequent oxidation, and hydrolysis. Between 10 and 30 min, the generation of hematite is approximately consistent with the disappearance of goethite, during which time leaching of the sulfides continues to generate ferrous iron at a rate greater than its oxidation and hydrolysis. Substantial oxidation and hydrolysis between 30 and 45 min results in rapid hematite formation and acid generation. This corresponds with rapid leaching of the nontronite, enhanced release of aluminum, and consequent formation of the sodium alunite/jarosite phase, as discussed previously. The estimated composition of this phase from the unit cell parameters is $Na_{0.9}(H_3O)_{0.1}Fe_{1.15}Al_{1.85}(SO_4)_2(OH)_6$ which hosts ~5% of the total iron in the leach residue.

The mineralogy of the leach residue at the conclusion of the co-processing test with 12% w/w nickel concentrate and 28% w/w nickel laterite was checked for comparison with that described previously. Consistent with the lower mass ratio of concentrate to laterite, the final residue was found to contain 39% hematite and 12% alunite/jarosite, on a normalized basis, with estimated alunite/jarosite composition of $Na_{0.9}(H_3O)_{0.1}Fe_{1.1}Al_{1.9}(SO_4)_2(OH)_6$. These values compare with 45% hematite and 12% alunite/jarosite with similar composition and reflect the lower iron and higher aluminum contents of the feed blend used in the 12:28 than the 10.5:19.5 ratio test.

### 3.4. Divalent Metal Sulfate Solubility

The addition of magnesium sulfate to the leach liquor to mitigate the formation of basic ferric sulfate and promote the formation of hematite during the pressure oxidation of the low-grade nickel concentrate was described in an earlier study [17]. Examination of the solubility of magnesium sulfate as a function of temperature based upon published information suggested that this salt is expected to be insoluble even at the terminal condition of the reaction: 250 °C, final free acidity 102.8 g/L (0.849 molal), and magnesium concentration 35.1 g/L (1.17 molal). The potential to form kieserite scale is therefore high and this is exacerbated by the presence of other divalent metal sulfates that include Ni, Co, Cu, Fe, and Mn, due to the common ion effect. The formation of kieserite-nickel kieserite scales during the HPAL of nickel laterite ore, and the dissolution of this scale phase during autoclave

cooling, was noted some time ago by Queneau et al. [60]. The in situ formation of kieserite from high Mg-content (14.1%) saprolite ore and its dissolution behavior have also been demonstrated by X-ray diffraction studies [61]. There is also potential for kieserite-szomolnokite ($MgSO_4.H_2O$-$FeSO_4.H_2O$) solid solutions [62] to form during the current tests since the solubility of ferrous sulfate in sulfuric acid is of similar magnitude to nickel sulfate at high temperatures [63]. However, as the concentration of ferrous sulfate does not remain high due to ferrous iron oxidation, the following discussion is focused mainly on the nickel and magnesium sulfate concentrations of co-processing leach liquor.

In the current study high concentrations of magnesium and nickel sulfates were generated in the co-processing tests conducted: Ni 15.1–23.0 g/L and Mg 11.2–16.2 g/L, the former mostly due to leaching of the low-grade concentrate and the latter primarily from the nickel laterite blend selected for the study. Figure 10 shows the solubilities of kieserite ($MgSO_4.H_2O$) and nickel kieserite ($NiSO_4.H_2O$) at various temperatures as a function of sulfuric acid concentration based upon published fits to experimental data [64,65]. Subsequently, Liu and Papangelakis [66] have undertaken chemical modeling to consistently fit these solubilities, and the solubility data for the $Al_2(SO_4)_3$-$H_2SO_4$-$H_2O$ system, at high temperatures.

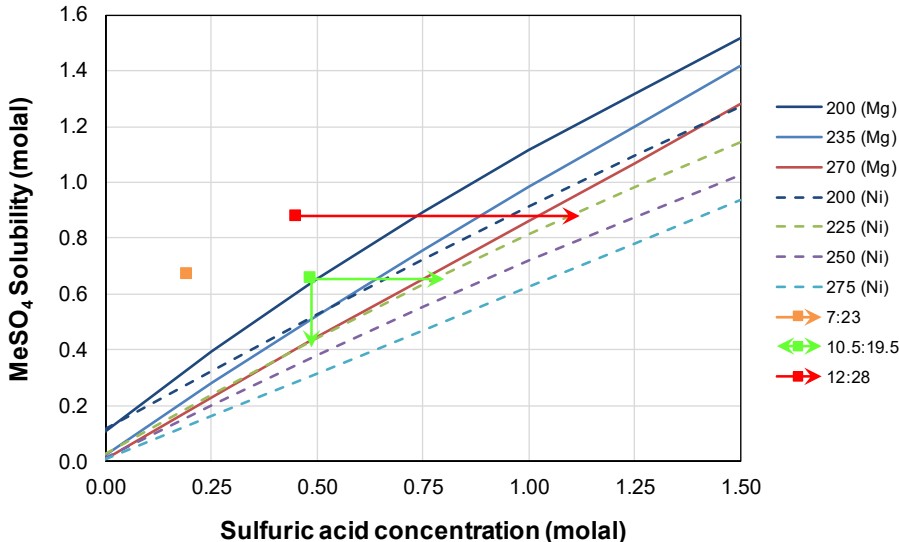

**Figure 10.** Modelled curves for the solubilities of $MgSO_4.H_2O$ and $NiSO_4.H_2O$ at selected temperatures (°C) as a function of sulfuric acid concentration [64,65]. Data points are for the final leach liquors obtained in the nickel concentrate to laterite tests in this study. The horizontal arrows selectively shown indicate the approximate sulfuric acid concentration required to keep the Mg and Ni in solution at 250 °C. The vertical arrow (selectively shown) indicates the approximate combined solubility of the nickel and magnesium sulfates at this temperature.

The data in Figure 10 clearly indicate that kieserite is more soluble than nickel kieserite for the range of temperatures and acid concentrations shown. The solubility of mixtures of these phases is expected to lie somewhere between the curves corresponding to the same temperature. The data points correspond to the sum of the nickel and magnesium sulfate concentrations in the final liquors and corresponding free acid concentrations. It is expected in each case that a significant amount of nickel and magnesium will precipitate as a solid solution of kieserite and nickel kieserite. The combined concentration of nickel and magnesium sulfates under leaching conditions can be simply and approximately calculated from the vertical line drawn from each data point to an estimated position between the 235 (Mg) and 250 (Ni) curves in Figure 10; this is shown for one of the points, i.e., the co-processing experiment employing 19.5% w/w nickel laterite with 10.5% w/w nickel concentrate. In order to simply and approximately calculate the acid concentration required to keep these metals in solution, a horizontal

line is drawn to an estimated position between the 235 (Mg) and 250 (Ni) curves, here shown for two of the co-processing experiments.

From the previous discussion, it is apparent that there is potential for scaling of the reaction vessel by metal sulfate salts when operated on a continuous basis (though these salts will dissolve when the temperature in the system is decreased [60,61], e.g., during scheduled maintenance). The combination of low-grade nickel concentrate and high-grade nickel laterite used in this study is therefore expected to result in reduced operating times, when compared to HPAL of the nickel laterite, before autoclave descaling is required.

There are several changes that could be implemented to mitigate the potential for salt scaling. The easiest and potentially most successful approach would be to use a nickel laterite blend having a lower magnesium content than used here, 3.32%, and preferably one with high limonite zone content, i.e., low in clay and serpentine minerals. Along similar lines, the use of a lower grade nickel concentrate will lead to lower nickel loading of the leach liquor and, may therefore provide greater sulfuric acid generating capacity if the sulfide content is similar, i.e., contains more iron sulfides. The up-front addition of acid or acid recycled from downstream processing may also assist, particularly where the acid generating capacity of the iron sulfides is low. Where the iron sulfides content is enough to provide the required acid, additional acid in the system can favor the formation of basic iron sulfate phases, i.e., alunite/jarosite and basic ferric sulfate, unless the acid is sufficiently well buffered. This could represent a dilemma in establishing the best process conditions, as acid buffering relies on there being sufficiently high concentrations of divalent metal sulfates present in the liquor. Balancing of the acid available from the different sources to ensure buffering is effective and yet minimizes the formation of basic iron sulfate phases therefore needs to be carefully assessed. Clearly, each combination of nickel sulfide and laterite ore will differ in its co-processing behavior and there will be an optimum blend, supply of additional acid, and set of process conditions that maximizes nickel and cobalt recoveries, ensures minimal losses of acid to the residues, and minimizes the potential for scaling by divalent metal sulfates. Optimizing high temperature process operation is not a trivial exercise as indicated by the experience of the Taganito HPAL plant project [67].

In summary, it is expected that a range of combinations of nickel-containing sulfidic materials and nickel laterite ores can be used to enable high extractions of the metal values. For each selected combination, it will be necessary to ensure that enough sulfuric acid (up-front added and in situ generated) is available to leach the laterite component and minimize divalent metal sulfate salt formation without promoting the formation of basic ferric sulfate phases.

### 3.5. Long Terms Storage and Oxidation of Residue Samples

The residues obtained from co-processing tests with 7:23 and 12:28 nickel sulfide to nickel laterite blends were stored after drying in plastic, screw-cap containers and their mineralogy was not examined until several years afterward. During this period, however, significant "solid state" transformations were noted to occur from XRD analysis, rendering any quantitative determination of mineralogy meaningless with respect to the tests per se. For selected samples the results obtained were distinctly different. The t0 sample from the 12:28 ratio test contained lesser amounts of all sulfide minerals, with the generation of hydronium jarosite $((H_3O)Fe_3(SO_4)_2(OH)_6)$, nickel hexahydrite ($NiSO_4.6H_2O$), and sulfur being noted; goethite may also have been generated but was already present in significant amounts to start. In comparison, the t10 sample did not contain sulfur.

Samples generated during the pressure oxidation of the low-grade nickel concentrate also gave varying outcomes. The t0 sample also contained lesser amounts of all sulfide minerals with generation of hydronium jarosite, nickel hexahydrite, sulfur, and goethite; the t10 sample contained only trace nickel hexahydrite, no sulfur, minor butlerite ($Fe(OH)SO_4.2H_2O$), and szomolnokite ($FeSO_4.H_2O$), along with major amounts of melanterite ($FeSO_4.7H_2O$) and copiapite ($AFe(III)_4(SO4)_6(OH)_2.20H_2O$, A = Fe(II), and/or Ni(II)). Possible mechanisms for the formation of some of these products have been discussed previously [17]. It must also, however, be noted that the samples were first prepared by

micronizing in absolute ethanol and the milled solids left to dry in the open atmosphere at ambient temperature. This may have facilitated some oxidation of the residual sulfide minerals and potentially impacted both the metal sulfate phases formed and their degree of hydration as ethanol will pick up water from the air. Notwithstanding this possibility, the sulfate salts generated have all been detected in Acid Mine Drainage evaporites and therefore represent oxidation products formed from iron- and nickel-containing sulfide minerals.

### 3.6. Comparison with Previous Studies

In the co-processing batch test work undertaken by Quinn et al. [3], a 70:30 laterite:sulfide blend was milled to form 28% w/w slurry in heap leach liquor and reacted at 220 °C with 900 kPa $O_2$ overpressure, enabling more than 90% Ni and Co extractions after 60 min, rising to 97.5% Ni and 96.4% Co after 250 min. These extractions and the associated discharge liquor assays do not differ greatly from the ranges encountered for these elements in the present study as shown in Table 3. The data shown for the current study were for the co-processing of 19.5% nickel laterite with 10.5% nickel concentrate, i.e., a 65:35 laterite:sulfide blend. Also shown in Table 3 are pilot test work data reported by Quinn et al. [3], which is discussed below.

**Table 3.** Comparison of the feed blend composition (% w/w), process liquor compositions (mg/L), free acid (g/L), and extraction data between the current study and that of Quinn et al. [3].

| Study | BlendSolids (% w/w) | | | | | | | | | |
|---|---|---|---|---|---|---|---|---|---|---|
| | Ni | Co | Fe | Mn | Al | Cr | Mg | Ca | Si | S |
| Current | 4.07 | 0.133 | 31.8 | 0.310 | 1.72 | 0.618 | 2.39 | 0.257 | 11.8 | 10.6 |
| Quinn et al. (B) | 1.24 | 0.055 | 12.2 | nr | 4.74 | nr | 5.46 | nr | nr | 3.20 * |
| Quinn et al. (P) | 1.41 | 0.020 | 15.4 | 0.072 | 3.72 | nr | 4.54 | 1.03 | nr | 2.52 * |

| Study | Process Liquor in (mg/L) | | | | | | | | | FA (g/L) |
|---|---|---|---|---|---|---|---|---|---|---|
| | Ni | Co | Fe | Fe(II) | Al | Cr | Mg | Ca | Na | |
| Current | <0.2 | <0.2 | <0.2 | <0.2 | <0.2 | <0.2 | 148 | 51 | 1380 | Nil |
| Quinn et al. (B) | 4350 | 326 | 41,500 | 1790 | 8550 | 834 | 21,300 | 371 | nr | 15.3 |
| Quinn et al. (P) | 3820 | 314 | 37,500 | 1540 | 7990 | nr | 15,900 | nr | 21,100 | 24.6 |

| Study | Process Liquor out (mg/L) | | | | | | | | | FA (g/L) |
|---|---|---|---|---|---|---|---|---|---|---|
| | Ni | Co | Fe | Fe(II) | Al | Cr | Mg | Ca | Na | |
| Current | 17,100 | 637 | 4820 | 1340 | 435 | 101 | 11,200 | 635 | 664 | 54.9 |
| Quinn et al. (B) | 13,000 | 669 | 5620 | nr | 2510 | nr | nr | nr | nr | 56.2 |
| Quinn et al. (P) | 8700 | 394 | 6710 | 2280 | 968 | nr | nr | nr | 880 | 46.7 |

| Study | Extraction (%) | | | | | | | |
|---|---|---|---|---|---|---|---|---|
| | Ni | Co | Fe | Mn | Al | Cr | Mg | Ca |
| Current | 98.0 | 97.8 | 3.3 | 96.0 | 5.4 | 3.4 | 89.3 | 61.8 |
| Quinn et al. (B) | 97.5 | 96.4 | nr | nr | nr | nr | nr | nr |
| Quinn et al. (P) | 91.9 | 78.8 | nr | nr | nr | nr | nr | nr |

B = Batch test data; P = Pilot test data; nr = not reported; * Corresponds to $S^{2-}$ content; FA = free acid; Pulp density of feeds (% w/w): Current 30, Quinn et al. (B) 28, Quinn et al. (P) 30.9–32.7.

Pilot testing revealed somewhat different behavior, as this employed a slightly different blend which was milled to form a higher pulp density slurry, 30.9–32.7% w/w, and although using similar composition heap leach liquor, the free acidity of this was adjusted to 24.6 g/L with sodium hydroxide prior to use. During testing the residence time was shorter than used for batch test work, 103–108 min, while a similar target temperature, 220 °C, and $O_2$ overpressure, 800–1000 kPa, were used. The high level of sodium, 20,100 mg/L, in the leach liquor will have provided a significant driving force for the formation of not only sodium alunite but also sodium jarosite solids, likely forming sodium

alunite/jarosite solid solutions, as the sodium concentration in the discharge liquor was significantly lower. In the initial stage of piloting, low nickel and cobalt extractions of 71.1% and 47.4%, respectively, were obtained. Various factors that include the higher pulp density, along with its negative impact upon the rate oxygen mass transfer, lower available sulfide content, significantly less complete ferrous iron oxidation (and hence iron hydrolysis) that is expected to improve at longer residence times, and the expected loss of acid to sodium alunite/jarosite solids resulted in significantly less acid being generated and/or available in situ. This was also reflected in the final free acidity of 21.1 g/L. Consequently, the second stage of the pilot study employed supplementary sulfuric acid addition of 150 kg/t blend, and this boosted the nickel and cobalt extractions to 91.9% and 78.7%, respectively (see Table 3). It also enabled the extent of ferrous oxidation to increase, and produced a final free acidity of 46.7 g/L.

The primary concern from both the batch and pilot test work (and this is confirmed from calculations using the analyses provided for the discharge liquors) is the concentration of divalent metal sulfates, in particular those of nickel and magnesium, and their potential to precipitate as monohydrate salts and form scale on the reaction vessel surfaces at sufficiently low free acid levels. Although not specifically noted by Quinn et al. [3], it is likely that metal sulfate scaling occurred; however, given that the pilot testing ran for eleven days, this may not have been a significant operational issue on that time frame. Although it is possible to form thick kieserite-nickel kieserite scales during HPAL test work [60], Liu et al. [68] have reported relative low amounts of scale formation from magnesium-rich laterite ore. In the latter study it was proposed that precipitated magnesium sulfate envelopes the hematite and alunite phases reducing their ability to scale. It was also noted by these authors that increasing slurry density and decreasing agitation speed (in addition to time) both exacerbate scaling.

Ferron and Fleming [7] initially demonstrated that elemental sulfur could be oxidized using low oxygen overpressure, 175 kPa, with no requirement for the addition of surfactants such as lignosol at 250 °C to effectively leach nickel and cobalt from a nickel laterite ore. Co-processing of a tropical laterite (% Ni 1.36, Co 0.110, Fe 42.3, Al 3.98, Cr 0.876, Mn 1.54, Mg 1.54, Si 4.18) with pyrrhotite (% Ni 0.94, Co 0.012, Cu 0.09, Fe 46.1, Al 1.75, Mg 1.15, Si 6.09, $S^{2-}$ 25.5) to give 26% w/w solids (presumably in tap water) was also conducted at 250 °C with 690 kPa oxygen overpressure for 120 min. Based upon its sulfide content, the pyrrhotite had an acid generating capacity of 763 kg/t (after correction for the base metal contents), marginally less than that of the low-grade nickel concentrate used in this study, ~800 kg/t. Equivalent acid additions of 256, 415, and 560 kg/t corresponding to estimated pyrrhotite to nickel laterite ratios of 0.34, 0.54, and 0.73, respectively, were tested, and while all additions gave high nickel and cobalt extractions. The highest addition also resulted in high aluminum extraction of ~50% and presumably, although not reported, high associated final free acidity. The study confirmed that 95%–97% of the sulfide content was oxidized; in the present study the extent of sulfide oxidation was not determined, though based upon the results obtained, is expected to be of similar magnitude.

## 4. Conclusions

The co-processing of a low-grade nickel concentrate with blended Bulong nickel laterite ore was demonstrated to be effective for the extraction of nickel and cobalt at 250 °C using total pulp densities of 30% w/w and 40% w/w over a range of nickel concentrate to nickel laterite mass ratios between 0.30 and 0.53. The use of low oxygen partial pressures to slow the reaction, combined with examination of the mineralogy using QXRD analysis, enabled a comprehensive understanding of the process to be obtained. The other specific findings from this study are as follows:

- High iron sulfide content feeds are highly suitable for co-processing with oxidic nickel-containing materials. Not only do these enable in situ sulfuric acid generation but the nickel and cobalt contents of the relevant minerals, typically pyrrhotite and/or pyrite. can be accessed;
- The low-grade nickel concentrate employed in this study had enough acid generating capacity to leach the nickel laterite ore without the need to provide supplementary acid;

- The mass ratio of nickel concentrate to nickel laterite can be tailored to ensure high base metal extractions and final free acidity, though the potential for the precipitation of divalent metal sulfates such as kieserite and nickel kieserite also needs to be minimized;
- Examination of the mineralogy of leach residue samples indicated that the oxidation sequence for the nickel and iron sulfide minerals is the same as that found when the nickel concentrate alone is leached; and
- For the tests conducted in this study, the iron hydrolysis products consisted mainly of hematite and an aluminum-rich sodium alunite/jarosite phase that hosts ~5% of the hydrolyzed iron in the leach residue.

The study indicated that the selection of the nickel concentrate and nickel laterite sources is expected to be important in establishing the feasibility of a co-processing approach for the extraction of metal values. Feed materials with low soluble magnesium and/or nickel contents are more likely to be amenable to co-processing, though this needs to be confirmed by appropriately designed test work to identify suitable process conditions that include temperature, oxygen partial pressure, pulp density, ratio of sulfide to laterite, process liquor composition, and supplementary (if any) acid addition.

**Author Contributions:** Conceptualization, methodology, investigation, data analysis, writing—original draft preparation, R.G.M.; Data analysis, writing—reviewing and editing, J.L. All authors have read and agreed to the published version of the manuscript.

**Funding:** Part of this research was funded by the Goldfields Esperance Development Commission through the Western Australian Government Royalties for Regions Regional Grants Scheme.

**Acknowledgments:** Member of the CSIRO analytical team, Milan Chovancek, Tuyen Pham, Bruno Latella Sophia Surin and Peter Austin are each thanked for their contributions to providing the analytical and XRD data.

**Conflicts of Interest:** The authors declare no conflict of interest. The funders had no role in the design of the study; in the collection, analyses, or interpretation of data; in the writing of the manuscript, or in the decision to publish the results.

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
