# Peer review of "The High Temperature Co-Processing of Nickel Sulfide and Nickel Laterite Sources"

_minerals, doi:10.3390/min10040351_

Round 1
Reviewer 1 Report
The paper is very interesting, and I am sure that it will attract may readers working on this subject. I can see good experimentation and reliable results. I congratulate the authors.
The paper is written very well and needs little correction/revision. Please see the attached file for my corrections/comments.
The paper can be accepted after minor revision to be published in the journal of Minerals.

Reviewer 2 Report
In the present manuscript, the authors McDonald and Li have conducted a high temperature co-processing of low-grade nickel sulfide concentrate with the high-grade laterite blend. Although the scheme is not new and well-known but within the interests of the readers of this journal. The manuscript is well organized and sufficiently provides the supportive data to prove the research hypothesis. However, this reviewer has several queries and comments on the manuscript, which must be addressed before its acceptance for the publication. They are:
- The abstract is always a stand-alone part of the manuscript, hence, it should adequately provide the overall picture of the entire work and key results. In this context, the authors should re-write the abstract with inclusion of relevant data, key findings, and the main conclusion drawn by the study.
- Line 21: use more suitable scientific terminology for the used word “sweetened”.
- In the introduction section, the authors have sufficiently discussed about the processing of nickel sulfide ores, however, the study also belongs to laterite processing. Hence, the discussion on laterite processing is worthy to include. For this, the authors may go through the recently published article: https://doi.org/10.1016/j.seppur.2019.115971
- The authors should add a note on the selection of types of materials. For example, why only the high-grade nickel laterite blend was chosen to mix with low-grade sulfide concentrate? What if, the combination will be changed? Do the authors see any adversity on the process when the combination will be changed to high-high or, low-low?
- Line 88: please mention the specification of the material of titanium vessel used in this study.
- Line 90-92: provide supportive reference.
- What was the moisture contents of the both samples in Table 1?
- In figures, the changes occurring for the free acid in the system is confusing at one sight. The representation from the starting point (zero time) will give more clarity to understand. Accordingly, the authors should modify the figures.
- It is strongly recommended to provide the morphological pictures of the mineral surfaces (SEM-EDX) while discussing about the phase change behaviour and jarosite precipitation. This can give an insight of the in-situ happening in the closed system.
- The authors should add descriptions on the calculations methodology for expressions use in this study.
- Line 190: what it means of “solid solutions”?
- Line 316-322: The authors have mentioned about the contribution of individual minerals for nickel and cobalt. The authors should give some quantitative data of both minerals dissolution in terms of metals liberation.
- Section 3.6: it will be better to summarize the key features of comparisons in a tabular form.
- Line 452: Correct “piloting” as “pilot”
- Line 470-476: can the authors provide the thickness of scaling and the difference of scaling thickness between the batch and pilot run?
- Line 490: it will be better if the data could be provided.
Line 492-496: This discussion is irrelevant in the context of this study.
